# OpenReview forum: "Verbalized Sampling: How to Mitigate Mode Collapse and Unlock LLM Diversity"
_ICLR.cc/2026/Conference — Submitted to ICLR 2026_

### Official Review · Reviewer_Kajm · 2025-10-24

**Soundness:** 2
**Presentation:** 3
**Contribution:** 3
**Rating:** 4
**Confidence:** 3

**Summary:**

This paper introduces a simple method, called verbalized sampling, to improve LLM diveristy. It's simple—get a model to generate responses _with probabilities over the answers_, and then pick from that according to some metric. They show that this improves diveristy at a given quality level. The submitted version has some issues that I'd like to see resolved, but overall I think this could be a nice paper, if the issues are resolved.

**Strengths:**

- Nice to have clearly defined definitions in Table 1, which I found to be very clear. Nice.
- I found Figure 2 to be well presented and clear, and I was happy to see the diversity and quality score here.
- Method is very simple, which I think is great, and its great to study the probability threshold and scaling.
- I think diversity with fixed quality is important, especially for synthetic data scaling. Indeed, I found the setting in section 7 compelling and useful.

**Weaknesses:**

- the analysis in section 3.1, using a linear reward model is interesting but I believe it may have a flaw. The probability under the base reference model does _not_ just capture typically, but will capture many features, such as correctness, etc. Therefore, i'd like to see additional analysis for 3.1. in particular, what % of time in preference data is the more typical response, using BOTH a response with higher log-likelihood under pretrained model AND a simple prompted model preferred. You could use a simply logistic regression feature approach, similar to the Sharma et al Sycophancy paper. You could do this for example on some other PM datasets too, which would be helpful.
- Moreover, you make claims about what _humans_ prefer, but in fact, _many_ humans are involved in the RLHF process, and it's unclear that they will all think the same text is typical. E.g., compare US and British english speakers. And the analysis is about the reward model scores, _not_ what humans actually prefer. Therefore, you should do the analysis I suggest above to isolate these questions. There are thus two questions: (i) what do humans prefer; (ii) what do PMs prefer. Different questions, worth studying individually, and you should not conflate the results of each type of analysis.
- there are some extra baselines I'd like to see:
    - I'd like to see the effect of the method compared to a pretrained model directly. How much of the diversity is actually recovered?
    - i'd like to see an input seeding example. e.g., "pull a random document from this set of 100" and then do the task. This is environmental randomness.
    - it is exactly right to look at diveristy, quality parteo curves. I suggest making this the main figure, and including it in the main text for Fig 2, **across all types** of task.
    - I'd like to see an additional quality check, which uses e.g., LLM as a judge and computes win-rates against a reference set, rather than just using rubrics.
- The writing is technically imprecise at times. small notes, as examples of this are below. In general, I'd like to see the strength of claims toned down and precision improved.
    - intro line 65: please crispy define typicality.
    - intro line 95: why is it principled? please explain crisply in text.
    - line 206 "to recover the diversity level". does this actually rcover it?
- I want to understand why the verbalization of probability distribution is helpful here? Do you have intuition around this? Can you explain this in the text?

**Questions:**

also, for Table 2, do you have a human assessment of quality for VS-standard vs e.g., direct?

(see weaknesses for most questoins)

---

> ### Author Response · Authors · 2025-11-25
> **Official Comment by Authors**
>
> We thank the reviewer for the constructive and encouraging feedback, especially the comments acknowledging our writing is clear and the results are compelling.
>
> > (W1) The analysis in section 3.1, using a linear reward model is interesting but I believe it may have a flaw...
> > (W2) ... There are thus two questions: (i) what do humans prefer; (ii) what do PMs prefer.
>
> We appreciate the suggestion to quantify "% of time the more typical response is preferred by both base and aligned model," as it can strength our emprical results and answer W2. We are finalizing the experiments and will update the manuscript with findings in few days.
>
> We also agree base model probability ($\log \pi_{\text{ref}}$) may capture features like correctness and fluency, not just typicality. **We did the controlled experiment in App. D.2 and further clarified in General Response (Concern 1).** To isolate typicality from quality, we utilized HelpSteer dataset, with human ratings for correctness and final reward. By using the logistic regression via the Bradley-Terry model (similar to Sharma et al) on 6,874 response pairs with **identical correctness ratings**, we found that human systematically prefer the typical response ($\alpha$: 0.57-0.65), confirming typicality bias exists independently of quality.
>
> > (W3.1) I'd like to see the effect of the method compared to a pretrained model directly...
>
> **We actually address this in Appendix F.1 (Fig. 13)**. We quantified diversity across Tulu-3 training stages (Base → SFT → DPO → RLVR). We observe that while SFT and RLHF cause severe diversity collapse in Direct, VS best recover the diversity against this collapse. By the final stage, VS retains **66.8%** of the Base Model's diversity, while Direct retains only **23.8%**.
>
> |Training Stage|Direct|VS|
> |:-|:-|:-|
> |Base (Llama-3-70B)|45.4|45.4|
> |Post-SFT (Tulu-3-SFT)|20.8|32.5|
> |Post-RLHF (Tulu-3)|10.8|**30.3**|
>
> > (W3.2) see an input seeding example.... environmental randomness
>
> Excellent point! **We performed the experiment of "Input Seeding"** by injecting random human-written examples from original datasets (environmental randomness) into the context, comparing Direct, Direct (+Env. Randomness), and VS. While this reduces the mode collapse of Direct prompting, VS consistently outperforms it across all tasks. This shows that the performance of VS stems from its active, structured search for diverse outputs, rather than simple stochasticity or random context variations.
>
> |Method|Poem Div. (↑)|Story Div. (↑) | Joke Div. (↑)|
> |:-|:-|:-|:-|
> |Direct|11.1±1.0|23.0±4.5|22.3±4.5|
> |Direct (+Env. Randomness)|14.5±1.8|27.8±5.2|44.4±3.9|
> |VS-Standard|**20.7±5.7**|**32.4±6.2**|**60.0±2.4**|
>
> > (W3.3) ... diveristy, quality parteo plot across all types of task.
>
> Great suggestion! We will feature the Pareto plot for creative writing in the main and add the rest to appendix. The trend is similar across tasks:
> * For dialogue simulation, VS-Standard approaches Pareto front balancing human-like donation amount distribution (diveristy) with human-like readability (quality).
> * For synthetic data, VS-Multi/CoT push the frontier by achieving higher downstream accuracy (quality) while maintaining higher diversity in synthetic data (diveristy).
>
> > W3.4 & Q1: Additional quality check & Human Study on Quality
>
> **We conducted human study combining pairwise comparison with win-rate analysis**. Specifically, we recruited 30 annotators per task to evaluate 90 pairs comparing Direct, Sequence, and VS-Standard on task-specific metric. **VS-Standard showed comparable win-rates against baselines on quality while achieving higher diversity.** Please **see our discussion of Concern 3 in General Response**.
>
> > (W4) The writing is technically imprecise at times. ... examples of this are below.
>
> Thanks. We'll take a full pass to strengthen the writing precision: specifically, defining "typicality" as the human tendency to prefer text that is familiar, fluent, and predictable based on cognitive psychology in Sec. 3.1, and changing "principled" to "theoretically grounded" as we intended to convey VS is grounded on theory rather than a heuristic. Regarding "recover," our response to W3.1 confirms VS indeed recover 66.8% of the diversity of base model.
>
> > (W5) I want to understand why the verbalization of probability distribution is helpful here?
>
> We will explain our intuition below and **provide detialed discussion in General Response (Concern 4)**. Standard prompts implicitly trigger mode-seeking behavior in aligned models: output one most typical response. By explicitly asking for probabilities over a set, we reframe the task from "ranking" to "sampling." This shifts the Representativeness Heuristic [1]: a typical list is ranked and contains bestsellers; a typical sample contains randoms item from across the distribution.
>
> [1] Kahneman, D., & Tversky, A. (1972). Subjective probability: A judgment of representativeness. Cognitive psychology, 3(3), 430-454.

---

### Official Review · Reviewer_oDEZ · 2025-10-30

**Soundness:** 1
**Presentation:** 4
**Contribution:** 2
**Rating:** 2
**Confidence:** 3

**Summary:**

The paper argued that one important cause of the mode collapse is the biased preference data. The authors investigated this “typicality bias” that once quantified is shown to be detrimental to model output diversity.
The authors propose a new prompt based method to bypass mode collapse, specifically by prompting models to list different response candidates in JSON while also listing their corresponding probability

**Strengths:**

- Overall, a solid and refreshing investigation into typicality bias, which identifies a key potentially undesired property in LLM preference data, and I believe this has great potential to unlock more LLM diversity.
- For each configuration of verbalized sampling, the experiment is thorough with good coverage of models, domains, varying levels of interactiveness, metrics, and human annotations.

**Weaknesses:**

- The motivation of the paper and the method feel mismatched — reading the paper feels like reading two. Typicality bias was thoroughly investigated and validated, but instead of continuing investigating any de-typicalized reward pairs, the authors used an unconvincing (see next point) reward-irrelevant method instead. The connection of typicality bias and verbalized sampling is at best, as the extensive appendix tried to prove under strong assumption, orthogonal. Line 203 to 207 (roughly) is not a coherent explanation either.
- Unsatisfactory baselines. The method used in the paper claims that requesting the distribution in the prompt is the key to improve observed diversity. However, strictly comparable list-based counterparts to "VS-*" methods probability request is subtly missing. To break it down Table 1:

| VS (k>1)          | List Level (k>1)  | Instance Level (k==1) |
|-------------|-------------|----------------|
| `VS-Standard`  | `Sequence`  | `Direct`       |
| `VS-CoT`        | **null** | `CoT`          |
| `VS-Multi` | **null** | `Multi-Turn`*  |

*`Multi-Turn` is an unfaithful and unacceptable misnomer. Unlike `VS-Multi` it only samples one more candidate per turn / call. So it is de facto instance level.

The missing comparable counterparts, which could be named `Sequence-CoT` and `Sequence-Multi`, are essential in evaluating the proposed method’s efficacy. Prior works and this one have shown that list does increase diversity over instance-level.

**Questions:**

- Line 178 claims "Having confirmed typicality bias". This claim lacks a description and content of the domains and types of the data. No description other than the dataset name is mentioned, yet the authors did not specify any conditions where the conclusion "human raters are biased towards responses more typical for the base model" are based on. So what are the domains of HelpSteer? How generalized can your claim be? Description about Eq1 lacks such context.
- For `α > 0 means that, holding the true utility fixed, higher typicality bias increases the reward`, it seems that some styles might just be universally preferred by users that the base learned to prefer already, which might drive up `α`. What could be a baseline `α` if not 0?
- For Creative writing evals, why `text-embedding-3-small` and `Claude 3.7`? Some explanation can be helpful.
- Double-check line 243. Not very readable.

---

> ### Author Response · Authors · 2025-11-25
> **Official Comment by Authors**
>
> We thank the reviewer for the rigorous feedback and for acknowledging our investigation into typicality bias as "solid and refreshing," and our experiment is "thorough with good coverage of models, domains." We will address your concerns on baselines and motivation below.
>
> > (W1) The motivation of the paper and the method feel mismatched ... The connection of typicality bias and verbalized sampling is at best, ... orthogonal.
>
> Excellent question! **To clarify: rather than orthogonal, VS restores diversity by exploiting the same typicality bias causing mode collapse, as detailed in General Response (Concern 4).** This leverages the Representativeness Heuristic [1]: humans judge a sequence as more typical of a random one than a repetitive one. We empirically validate that LLMs mimic this preference. So models are able to mode-collapse to a "typical" diverse sample when prompted with VS. De-typicalized reward pairs is also very interesting idea. However, de-typicalization may be really challenging given the fundamental and pervasive nature of the typicality bias. We propose that VS might serve as a baseline for the exploration, as it provides an immediate, inference-time intervention that works on existing models.
>
> [1] Kahneman, D., & Tversky, A. (1972). Subjective probability: A judgment of representativeness. Cognitive psychology, 3(3), 430-454.
>
> > (W2) Unsatisfactory baselines. However, strictly comparable list-based counterparts to "VS-*" methods probability request is subtly missing.
>
> We thank the suggestion for additional baselines. To ensure fair evaluation, **we have implemented `Sequence-CoT` and `Sequence-Multi` as strictly comparable list-based baselines** on creative writing (Poem/Joke) and synthetic data task.
>
> |Setting|Poem Div. (↑)|Joke Div. (↑)|Math Acc. (↑)|
> |:-|:-|:-|:-|
> |Sequence|17.3±6.5|55.2±3.7|34.3|
> |Sequence-CoT|18.4±6.3|57.4±2.9|33.6|
> |Sequence-Multi|19.5±7.7|57.2±2.1|34.3|
> |VS-Standard|20.7±5.7|60.0±2.4|36.1|
> |VS-CoT|24.3±6.1|60.4±2.6|36.9|
> |VS-Multi|**24.8±7.5**|**60.5±1.7**|**37.5**|
>
> **VS variants consistently outperform Sequence counterparts, with even simpler VS-Standard outperforms the more complex Sequence-Multi across tasks**. Notably, adding CoT to Sequence decreases synthetic task performance (33.6 vs 34.3). **Please see our discussion of Concern 2 in General Response above.**
>
> **We respectfully argue that Multi-turn is still list-level.** While it generates one response per call, it conditions on the history of previous outputs to encourage diversity. Truly instance-level methods (Direct, CoT) possess no memory of prior attempts and generate one response/call.
>
> > (Q1) So what are the domains of HelpSteer? How generalized can your claim be? Description about Eq1 lacks such context.
>
> HelpSteer contains 21k samples from real-world ChatGPT conversations (ShareGPT) covering diverse use cases, and proprietary prompts (e.g. summarization, closed QA, and extraction). To verify generalizability beyond HelpSteer, we validated across three additional datasets and five base models (App. D.1), finding consistent trends. Main purpose of Eq1's is to isolate typicality from correctness: App. D.2 shows that even on correctness-matched pairs, humans prefer typical responses 57–65% of the time, confirming the bias exists independently of quality. **Please see our discussion of Concern 1 in General Response.**
>
> > (Q2) ... it seems that some styles might just be universally preferred by users that the base learned to prefer already. What could be a baseline α if not 0?
>
> Excellent question! It is true that some aspects might be encapsulated by base model probability (e.g., fluency). In our framework (Eq1), if $\pi_{ref}$ captures aspects of quality, we can simply interpret $r_{true}$ as the "leftover" utility, the quality aspects not already captured by the base model probability. Our proof remain valid: the typicality defined by the base model still causes mode collapse. Ideally, if humans treated base model probability purely as a neutral carrier for content, $\alpha=0$. Empirically, we observe $\alpha \approx 0.5-0.6$, confirming typicality acts as an independent, additive driver of preference.
>
> > (Q3) For Creative writing evals, why text-embedding-3-small and Claude 3.7?
>
> We followed established literature: text-embedding-3-small for diversity measurement [1] and Claude-3.7 for quality evaluation following the Creative Writing V3 Leaderboard [2], which offers optimal cost-performance). We will update the manuscript to clarify this in Section 5.
>
> [1] Zhu, Alan, et al. "BARE: Leveraging Base Language Models for Few-Shot Synthetic Data Generation." arXiv preprint arXiv:2502.01697 (2025).
> [2] https://eqbench.com/creative_writing.html
>
> > (Q4) Double-check line 243. Not very readable.
>
> Thanks. We will revise Line 243 and the surrounding section to improve flow and clarity.
>
> --
> We hope this addresses your concerns. Please let us know if you have further questions, we are happy to discuss further.

---

> > ### Comment · Reviewer_oDEZ · 2025-11-25
> > **Quick Questions**
> >
> > Before I address the other points, I want to get more clarifications on my key concern (I am nicely surprised by the results).
> > Specifically:
> > - Please paste all the prompts you used for `Sequence-Multi` and `Sequence-CoT`, either as comment or update it as PDF.
> > - Which model did you use in reporting these new experiments?
> > - is `k = 5` settings and are all other such hyper parameter settings the same (temp=.7, 8192 new tokens, top-p=1)? So everything is the same except the `Sequence-*` variants do not have prompts about probability? A 3% accuracy swing is rather significant.
> > - Do you intend to include these experiments in the main sections of your updated draft? (For this point, I just need to know your plan. I do not need to see the updated PDF for this during the review period since this can be too much rewriting to request.)

---

> ### Author Response · Authors · 2025-11-25
> **Thanks for your quick response, here's the clarification to your questions**
>
> > **Reviewer:** Please paste all the prompts you used for `Sequence-CoT` and `Sequence-CoT`...
>
> Below are the exact prompts used. We provide the `VS-*` prompt alongside as a reference.
>
> **1. CoT Setting**
> Both `Sequence-CoT` and `VS-CoT` generate multiple responses after the thinking process, but only VS-CoT includes probabilities for each response.
>
> * **Sequence-CoT**
>     ```python
>     Generate {num_samplings} responses to the input prompt.
>
>     First, provide a single "reasoning" field as a string, detailing your step-by-step thought process.
>     Then,  exactly {num_samplings} responses in "responses" field as a Python list of strings, formatted as:
>     ["response1", "response2", "response3", ...]
>     Return ONLY a JSON object with "reasoning" and "responses" fields, with no additional explanations or text.
>     ```
> * **VS-CoT**
>     ```python
>     Generate {num_samplings} responses to the input prompt.
>
>     First, provide a single "reasoning" field as a string, detailing your step-by-step thought process.
>     Then, return the output in JSON format with the key "responses" (list of dicts). Each dictionary must include:
>     - 'text': the response string only (no explanation or extra text).
>     - 'probability': the estimated probability from 0.0 to 1.0 of this response given the input prompt (relative to the full distribution).
>     Return ONLY the JSON object, with no additional explanations or text.
>     ```
>
> **2. Multi-turn Setting**
> `Sequence-Multi` requests a simple list, while `VS-Multi` requires a dictionary with probabilities:
>
> * **Sequence-Multi**
>     ```python
>     Generate {num_samplings} responses to the input prompt.
>
>     First, sample {num_samples_per_prompt} responses.
>     Return the responses as a Python list of strings, formatted as:
>     ["response1", "response2", "response3", ...]
>     Return ONLY the list, with no additional explanations or text.
>     ```
> * **VS-Multi**
>     ```python
>     Generate {num_samplings} responses to the input prompt.
>
>     First, sample {num_samples_per_prompt} responses.
>     Return the responses in JSON format with the key: "responses" (list of dicts). Each dictionary must include:
>     - 'text': the response string only (no explanation or extra text).
>     - 'probability': the estimated probability from 0.0 to 1.0 of this response given the input prompt (relative to the full distribution).
>     Return ONLY the JSON object, with no additional explanations or text.
>     ```
>
> Both `Sequence-Multi` and `VS-Multi` use this multi-turn structure. For turn $\geq$ 2, the user prompt to continue is as follows for both settings:
> ```python
> "Generate {num_samples_per_prompt} alternative responses to the original input prompt."
> ```
>
> **Qualitative Example**
>
> To provide a concrete example demonstrating why VS outperforms Sequence, we prompted Claude Opus 4.1 to 'Generate US states,' aiming to observe the elicitation of long-tail answers.
>
> * Sequence: https://imgur.com/a/L7NUiPy
>
> * VS: https://imgur.com/a/9sBKc4q
>
> The results show that VS successfully prompts the model to generate lower-probability states (e.g., Oregon and Wyoming), whereas Sequence continues to yield high-frequency states, such as California and Texas.
>
> ---
>
> > **Reviewer:** Which model did you use in reporting these new experiments?
>
> We reported the average results across four distinct models to ensure robustness: **GPT-4.1**, **Gemini-2.5-Flash**, **Claude-4-Sonnet**, and **Llama-3.1-70B**.
>
> ---
>
> > **Reviewer:** is $k = 5$ settings, and are all other such hyperparameter settings the same... A 3% accuracy swing is rather significant.
>
> Yes, we confirm that the generation settings are identical to the main experiments ($k=5$, temperature=0.7, max new tokens=8192, top-$p$=1).
>
> ---
>
> > **Reviewer:** Do you intend to include these experiments in the main sections of your updated draft?
>
> We plan to include this experiment as an ablation study, with the results and corresponding analysis included in the Appendix of the final version. We can also add mentions of this to the main paper at Section 5.1, and can make this update within the review period. As context, our concern is the impact of this presentation dimension on the readability of the already quite compressed Figure 4, methodology, etc. We hope this strikes a reasonable balance, but would welcome further feedback or other suggestions!

---

### Official Review · Reviewer_fHBG · 2025-11-01

**Soundness:** 3
**Presentation:** 4
**Contribution:** 3
**Rating:** 6
**Confidence:** 4

**Summary:**

* The paper identifies **typicality bias** in human preference data as a key cause of **mode collapse** in RLHF-aligned large language models.
* It shows theoretically and empirically that this bias sharpens the model’s distribution, reducing output diversity.
* The authors propose **Verbalized Sampling (VS)**, a simple, inference-time prompting method that restores diversity without retraining, achieving higher variety while maintaining quality.

**Strengths:**

1. **Novel framing**
   The paper introduces a fresh and compelling perspective by identifying *data-level human preference bias* (typicality bias) as a root cause of mode collapse. This shifts the discussion from algorithmic issues in RLHF to psychological factors in human annotations, offering a new concept for understanding mode collapse.

2. **Strong empirical evidence**
   The paper validates the typicality-bias hypothesis on real preference datasets such as HELPSTEER, showing consistent and statistically significant results across multiple base models. This empirical grounding gives credibility to the theoretical claims.

3. **Practical mitigation method**
   The proposed *Verbalized Sampling (VS)* is simple, training-free, and effective. It improves output diversity through prompt-level control at inference time, without modifying model weights. The method is orthogonal to existing decoding strategies and easy to adopt in practice.

4. **Strong and comprehensive empirical validation**
The experiments are comprehensive and convincing. The paper tests its hypothesis across multiple datasets, diverse task categories, and several subtasks.

**Weaknesses:**

1. **Limited validation of the core hypothesis**
    - The central assumption—that *human annotators prefer more typical responses*—is not directly validated.
    - The authors infer this pattern indirectly through reward model correlations. A direct analysis of whether humans explicitly favor more typical responses would make the hypothesis stronger.
2. **Inference cost and deployment feasibility**
    - While *Verbalized Sampling (VS)* effectively increases diversity, it requires generating *k* responses and verbalizing probabilities, leading to roughly *k×* higher inference cost. This makes the approach difficult to apply in large-scale LLM deployments despite its conceptual simplicity.
3. **Lack of qualitative analysis**
    - The paper would benefit from more qualitative examples to illustrate how VS improves diversity without hurting quality. Current evaluations are largely quantitative, leaving uncertainty about how these improvements appear in actual outputs.
    - In particular, there should be a qualitative comparison between generating responses individually (one-by-one) and generating multiple responses simultaneously (e.g., 5 at once). Although the paper reports quantitative gains, it remains unclear whether the perceived quality or linguistic characteristics of the responses differ between these settings. Providing human evaluations or illustrative examples would clarify whether VS changes not just diversity scores but the actual quality of the outputs.
4. **Missing ablation studies**
    - Figure 14 could also analyze how varying the number of generated candidates in a direct setting appears, and variants like VS-CoT or VS-Multi should also be compared in Figure 14. This would clarify which components of VS contribute most to the observed gains.
5. **Fairness issue in VS-Multi comparison**
- The comparison involving VS-Multi appears potentially unfair, as it effectively produces a larger number of responses (e.g., *5×k* candidates) than other methods. This difference may inherently boost diversity and performance, so comparisons should control for the total number of generated outputs across settings.

**Questions:**

- Q1 What is the expected inference cost for each of the proposed prompt settings?

- Q2 Considering cost, diversity, and quality, which approach appears to be the most feasible overall?

- Q3 How was the distribution of model-generated probabilities?
Did the distribution differ across different VS variants, and is there any observed relationship between those probabilities and the quality of generated outputs?

---

> ### Author Response · Authors · 2025-11-25
> **Official Comment by Authors**
>
> We sincerely appreciate reviewer's positive feedback, particularly for finding our perspective fresh and compelling, our empirical results strong, and our proposed method simple yet effective. We hope to clarify any concerns as follows:
>
> > (W1) The central assumption—that human annotators prefer more typical responses—is not directly validated...
>
> **We validated the hypothesis directly on human data in two steps (App D.1, D.2).** In App D.1, we calculate the "Typicality Bias Rate" across five base models on four preference datasets. On datasets with only human annotation, we found that human consistently preferred more typical responses above 50% (**51.6%-60.8%**), confirming the existence of typicality in human preference. In App D.2, we modeled final reward as a mix of utility and typicality (Eq1). Using 6,874 correctness-matched pairs from HelpSteer, we found humans significantly prefer responses with higher typicality even when correctness is identical (**$\alpha$ = 0.57-0.65**). **Please see our discussion of Concern 1 in General Response above.**
>
> > (W2, Q1, Q2) ...vs requires generating k responses and verbalizing probabilities, leading to roughly k× higher inference cost...
>
> We agree that understanding cost/diversity trade-off is critical and will answer this concern with Q1 and Q2. **We performed additional experiement/analysis on the poem generation (2000 responses with GPT-4.1, Claude-4-Sonnet)**. We measured total token consumption, API costs, and latency, comparing baselines (1 response/call) with VS (k=5 candidates). **Multi-turn is the most expensive strategy due to context accumulation, whereas VS-Standard remains highly efficient (1.12× cost)**.
>
> |Method|Cost ($)|Rel. Cost|Time per batch (s)|Rel. Time|Diversity|Div. Gain|
> |:-|:-|:-|:-|:-|:-|:-|
> |Direct|5.75±0.29|1x|2.53|1x|11.1±1.0|1x|
> |Sequence|6.38±0.27|1.11x|2.91|1.15x|17.3±6.5|1.56x|
> |Multi-Turn|7.48±0.45|1.30x|6.80|2.69x|14.1±2.3|1.27x|
> |VS-Standard|**6.42±0.32**|**1.12x**|**3.11**|**1.23x**|**20.7±5.7**| **1.86x**|
> |VS-CoT|8.68±0.43|1.51x|4.21|1.66x|24.3±6.1|2.19x|
> |VS-Multi|9.15±0.51|1.59x|7.12|2.81x|24.8±7.5|2.23x|
>
> **VS-Standard vs. Sequence.** With almost the same cost (1.12× vs 1.11×), VS-Standard achieves higher diversity (1.86× vs 1.56×), confirming gains stem from probabilistic guidance, not just token overhead.
>
> VS-Standard offers a highly favorable trade-off, exchanging modest 12% cost and 23% latency for 86% diversity gain. **This aligns with modern inference trends** (CoT, o1-reasoning trades inference time for quality) where spending marginal compute is accepted to unlock significantly better performance. For applications like creative writing or synthetic data, this exchange is economically reasonable.
>
> > (W3) Providing human evaluations or illustrative examples would clarify whether VS changes not just diversity scores but the actual quality of the outputs.
>
> To verify that diversity gains do not compromise quality, we **conducted additional human study on quality (joke, poem, story)**. **VS-Standard achieved comparable quality win-rates against baselines while maintaining higher diversity.** For qualitative examples, we will expand **current VS examples in Appendix G.4** with additional Direct vs. VS comparisons across all tasks (building on) in the updated manuscript. **Please see our discussion of Concern 3 in General Response above.**
>
> > (W4) Figure 14 could also analyze how varying the number of generated candidates in variants like VS-CoT or VS-Multi...
>
> Thanks for the suggestion. Appendices F.2/F.3 (Figs 14-17) already provide extensive ablations on how number of candidates, temperature, top-p, and min-p affect the pure VS-Standard performance. We agree that including VS-CoT/Multi variants would provide a more complete picture and will add this comparison in the revision.
>
> > (W5) The comparison involving VS-Multi appears potentially unfair...
>
> Thank you for raising this point. To clarify: **we controlled for total computational budget the same across all methods** (See Table 1). VS-Multi distributes the same budget $N$ across multiple turns.
>
> > (Q3) How was the distribution of model-generated probabilities? ...
>
> **We investigated verbalized probability distributions against pre-training distributions in Appendix E.9 (Fig. 12) and also compared across the VS variants.** VS-Standard’s verbalized probability closely aligns with the pre-training distribution (low KL), whereas Direct collapses to a few modes. Across Variants, VS-CoT exhibited the lowest KL Divergence.
>
> |Model/KL|VS-Standard|VS-CoT|VS-Multi|
> |:-|:-|:-|:-|
> |**Claude-4-Sonnet**|0.150|0.093|0.111|
> |**GPT-4.1**|0.139|0.126|0.132|
>
> We also observed a strong **functional relationship between the verbalized probabilities and output quality/diversity**. In Appendices F.5/F.6, we found that explicitly instructing the model to sample responses with probabilities below 0.1 (tail distribution) increases diversity while maintaining or even improving quality.

---

### Author Response · Authors · 2025-11-25
**General Response [1/2]**

We thank the reviewers for their constructive feedback and are encouraged that they recognized several **key strengths** of our work.

* **Novel Perspective:** Reviewers mentioned the identification of typicality bias as a "fresh and compelling perspective" (`fHBG`) and a "solid and refreshing investigation" (`oDEZ`) that offers new insights into mode collapse and diversity.
* **Simple yet Effective Method:** Verbalized Sampling (VS) was acknowledged for being "very simple" (`Kajm`), "training-free," and "orthogonal to existing decoding strategies" (`fHBG`), making it easy to adopt in practice.
* **Comprehensive Experiments:** All reviewers acknowledged the empirical rigor, describing the experiments as "comprehensive and convincing" (`fHBG`) and "thorough" (`oDEZ`). The specific analyses, such as the creative writing results and synthetic data tasks, were also highlighted as clear and compelling (`Kajm`).

We appreciate above positive feedback and would like to address several **common concerns** below.


### **(Concern 1) Validation of Typicality Bias in Human Preference**
We thank the reviewer `fHBG` and `Kajm` for raising the need to validate the assumption of "human tend to prefer more typical text" direct on human data (`fHBG`) and to control for confounding factors such as correctness alongside typicality (`Kajm`).

**We clarify that we have validated this hypothesis through three progressive steps in the paper:**

1. **Theoretical Grounding in Cognitive Psychology (Section 3.1):**
We first identify typicality bias based on established theory in cognitive psychology. Previous studies in cognitive psychology (e.g., the mere-exposure effect[1], processing fluency[2], schema congruity[3], etc) showed that humans tend to prefer text that is familiar, fluent, and predictable. We hypothesis that **these cognitive tendencies lead to a typicality bias in human preference data**.
3. **Empirical Verification on Human Data (Appendix D.1):**
We use the log probability from pretained base model ($\log \pi_{ref}(y|x)$) to approximate text typicality (Intuition: as base model has been trained to maximize likelihood on massive text corpora, it will assign higher probability to more typical text).
To confirm typicality bias exist in human preference data, in Figure 5, we **measured the "Typicality Bias Rate" across five base models on four preference datasets** (OpenAI TL;DR, UltraFeedback, HelpSteer, and Skywork Preference), which measures how often human annotators prefer responses with higher base-model probability. We specifically highlight OpenAI TL;DR and HelpSteer, which rely on pure human annotation. In both datasets, **human annotators prefer the "more typical" response (the one with higher base-model probability) consistently above chance (> 50%)**, ranging from **51.6% to 60.8%**. This confirms that typicality bias is a pervasive property of  preference data.
3. **Validation Controlling for Correctness (Appendix D.2):**
To address the concern that the responses are preferred simply because they are "correct", we isolated typicality from correctness using the **HelpSteer** dataset, which contains explicit instance-level human ratings for the final reward (overall helpfulness) and true task utility (correctness). We **analyzed 6,874 pairs of responses where human annotators rated the correctness as identical**. By modeling the final reward $r(x, y)$ as a combination of utility ($r_{true}$) and typicality ($\log\pi_{ref}$):
$$r(x, y) = r_{true}(x,y)+ \alpha\log\pi_{ref}(y \mid x)+ \epsilon(x)$$
We found that even when human-rated correctness ($r_{true}$) is fixed, **human annotators still systematically prefer the response with higher base-model probability**, with $\alpha$ ranging from **0.57 to 0.65**. This confirms that human raters are biased towards responses more typical (for the base model), independent of objective correctness.

References:

[1] Zajonc, R. B. (1968). Attitudinal effects of mere exposure. Journal of personality and social psychology, 9(2p2), 1.

[2] Reber, R., Schwarz, N., & Winkielman, P. (2004). Processing fluency and aesthetic pleasure: Is beauty in the perceiver's processing experience?. Personality and social psychology review, 8(4), 364-382.

[3] Meyers-Levy, J., & Tybout, A. M. (1989). Schema congruity as a basis for product evaluation. Journal of consumer research, 16(1), 39-54.

---

> ### Author Response · Authors · 2025-11-25
> **General Response [2/2]**
>
> ### **(Concern 2) Missing baselines**
> We appreciate the suggestion from Reviewer `oDEZ` for additional baselines. We have now implemented the suggested `Sequence-CoT` and `Sequence-Multi` as comparable list-based baselines.
>
> **Results: Creative Writing (Poem & Joke) & Synthetic Data Task**
> |Setting|Poem Diversity (↑)|Joke Diversity (↑)|Math Accuracy (↑)|
> |:-|:-|:-|:-|
> |Sequence|17.3±6.5|55.2±3.7|34.3|
> |Sequence-CoT|18.4±6.3|57.4±2.9|33.6|
> |Sequence-Multi|19.5±7.7|57.2±2.1|34.3|
> |VS-Standard|20.7±5.7|60.0±2.4|36.1|
> |VS-CoT|24.3±6.1|60.4±2.6|36.9|
> |VS-Multi|**24.8±7.5**|**60.5±1.7**|**37.5**|
>
> **VS variants consistently outperform all the sequence-level baselines**, achieving higher diversity in creative writing tasks and better downstream accuracy in synthetic data task. The above experiments isolates the effect of CoT reasoning and multi-turn interaction in VS, and confirms our method's effectiveness.
>
> **Explanation for original baselines:** The original baselines (Sequence[1], Multi-turn[2]) were selected to align with established work on diversity and creativity. VS-CoT/Multi are not designed to compete unfairly with these baselines; rather, they serve as practical variants to reduce the cognitive burden of generating multiple responses with probabilities simultaneously. These variants try to maintain comparable quality to instance-level prompt and diversity to VS-Standard.
>
> [1] Meister, N., Guestrin, C., & Hashimoto, T. B. "Benchmarking distributional alignment of large language models." arXiv preprint arXiv:2411.05403 (2025).
>
> [2] West, P., & Potts, C. (2025). Base models beat aligned models at randomness and creativity. arXiv preprint arXiv:2505.00047.
>
>
> ### **(Concern 3) Human Study on Quality**
> We thank reviewer `fHBG` and `Kajm` for raising the point that additional human evaluation of quality would complement the automatic evaluation and strengthen the paper.
>
> **We have performed additional human studies on quality** during rebuttal to verify that diversity gains do not compromise quality. We recruited 30 annotators per task to evaluate 90 pairs comparing Direct, Sequence, and VS-Standard on funniness (jokes), pleasantness (poems), and engagement (stories) using a 4-point Likert scale (A ≫ B to A ≪ B). The evaluation were conducted on the same 90 pairs used for diversity in Section 5.2.
>
> We report the win-rate across method pair for each task. **VS-Standard showed comparable win-rates against baselines on quality while achieving higher diversity.** Notably, while VS-Standard shows a slight quality drop on stories (0.455 vs. Direct), this aligns with our findings in Section 5.1 that complex prompts (VS) can create cognitive burden causing quality drop. IAA was moderate for stories (0.49), high for poems (0.64) and jokes (0.79).
>
> |Task|VS vs Direct|VS vs Sequence|Direct vs Sequence|
> |-|- |-|-|
> |Joke|0.547|0.644|0.617|
> |Poem|0.524|0.516|0.509|
> |Story|0.455|0.569|0.593|
>
> ### **(Concern 4) Connecting Typicality Bias and Verbalized Sampling**
>
> We thank Reviewer `oDEZ` and `Kajm` for highlighting the need to clarify our theory connection. We recognize our original writing was unclear and have substantially strengthened this link. We will update the paper accordingly, but the ideas are summarized below:
>
> **The key insight**: VS exploits rather than bypasses typicality bias through the Representativeness Heuristic (Tversky & Kahneman, 1972). Humans judge sequences as "typical" when they exhibit expected statistical properties. Crucially, this creates opposite effects by prompt type:
>
> - **Instance prompts ("Tell a joke"):** Typicality bias → single most probable response → mode collapse
> - **Distribution prompts ("Generate 5 jokes with probabilities"):** Typicality bias → statistically diverse sample → diversity preservation
>
> We provide additional empirical validation that LLMs inherit this cognitive bias. In controlled experiments testing sequence preferences (we plan to add it in Appendix D), models overwhelmingly judge diverse sequences as more "typical/representative" than repetitive ones (95-100%, p<0.001), despite equal probability under independence. This confirms our mechanism and aligns with recent work showing LLMs exhibit human-like probability judgments [1].
>
> Theoretically, we now formalize this with more relaxed assumptions. Given that humans prefer representative probability distributions at a rate better than chance, this breaks the flat-reward condition underlying collapse, enabling diversity recovery. Under sharpening ($\gamma>1$), VS converges to the typical set rather than modal points – recovering 66.8% of base model diversity versus 23.8% for direct prompting.
>
> The connection is thus fundamental: the same bias causing collapse for instances preserves diversity for distributions. We will revise sections 3-4 and Appendix D.4 provide complete formalization.
>
> [1] Zhu, J. Q., & Griffiths, T. L. (2024). Incoherent probability judgments in large language models. arXiv preprint arXiv:2401.16646.

---

> > ### Author Response · Authors · 2025-12-01
> > **Summary of the Update and New Revision of our Paper**
> >
> > Based on the feedback, we have updated our manuscript accordingly and uploaded a new version of our paper for review. The changes are highlighted in orange. We summarize the key changes:
> > * **Clarified the validation of the core hypothesis (human annotator prefer more typical responses) in Section 3.1** with further experiment detials in Appendix D.1-D.2, explicitly stating the validation process from empirical experiments on four preference datasets and regression analysis to disentangle typicality from correctness (Reviewer fHBG and Kajm).
> > * **Added a human study on quality in Section 5.2 and Appendix E.3**, alongside an expanded **Appendix H containing side-by-side qualitative comparisons between Direct and VS** across all tasks (Reviewer fHBG and Kajm).
> > * **Clarified and strengthened the theoretical link between typicality bias and Verbalized Sampling in Section 4** and Appendix D.3–D.8, formally proving how VS restores diversity by exploiting the same typicality bias causing mode collapse (Reviewers oDEZ, Kajm).
> > * **Added strict list-based baselines** (Sequence-CoT and Sequence-Multi) in **Appendix E.11** (Table 27), demonstrating that VS variants constantly outperform the Sequence counterparts (Reviewer oDEZ).
> > * **Added the ablation experiment on diversity across post-training stages in Section 5.1**, using the Tulu-3 family to empirically demonstrate how post-training causes mode collapse and how VS mitigates it (Reviewer Kajm).
> > * **Added an "Input Seeding" baseline (environmental randomness) in Appendix E.11** (Table 28), confirming that VS performance stems from its active, structured search for diverse outputs rather than random context variations (Reviewer Kajm).
> > * **Included a detailed Cost & diveristy/quality Analysis in Appendix G.2**, featuring a new table (Table 29) that quantifies the trade-off between inference cost, latency, and diversity gains (Reviewer fHBG).
> > * **Clarified computational fairness in Table 2**, highlighting the "same computation budget" constraint in the table caption (Reviewer fHBG).
> > * **Added Pareto-optimal analysis for all tasks in Appendix E.1** and Figure 7, illustrating the better diversity-quality trade-off of VS across tasks (Reviewer Kajm).
> > * **Added typicality bias in post-trained instruct models in Appendix D.1** (Reviewer Kajm).
> > * **Rephrased the line 243** for clarity (Reviewer oDEZ).
> > * **Took a full pass to strengthen the writing precision** and clearly defined the typicality in line 65 and tone down the "principled" in Line 95 (Reviewer Kajm).

---

### Author Response · Authors · 2025-12-03
**Note to AC (summary for rebuttal phase)**

Dear Area Chair,

We note the recent incident on OpenReview and sincerely appreciate your time and effort in reviewing our paper. In our final comment, we would like to provide a summary of the replies, the changes we have made, and the state of the discussion.

1. **All three reviewers found the problem formulation to be novel and the proposed method is effective:**
    * **fHBG:** Describes the work as a "fresh and compelling perspective" on typicality bias as the root of mode collapse. They note the empirical validation "gives credibility to the theoretical claims" and the method is "simple, training-free, and effective."
    * **oDEZ:** States the paper offers a "solid and refreshing investigation" with "great potential to unlock more LLM diversity," supported by "thorough" experiments across models, domains, and metrics.
    * **Kajm:** Finds the work "well presented and clear," the method "very simple," and the experimental setting "compelling and useful."

2. **We made changes to improve the paper. Major revisions include:**
    * **Direct Validation of Typicality Bias (Sec 3.1):** Addressing `fHBG` and `Kajm`, we incorporated App D.1/D.2 content into the main text. We explicitly demonstrate that humans prefer typical responses (>50% of the time) on preference datasets and provided a regression analysis proving this bias persists even when controlling for correctness.
    * **Theoretical Proof (Sec 4, App D.3–D.8):** Addressing `oDEZ`, we formally proved that Verbalized Sampling (VS) restores diversity specifically by exploiting the typicality bias that causes mode collapse in standard prompting
    * **New Baselines & Ablations:** Addressing `oDEZ` and `Kajm`, we implemented strict list-based baselines (Sequence-CoT/Multi), an "Input Seeding" baseline (App E.11), and moved Tulu-3 ablation to Sec 5.1 to empirically demonstrate how post-training causes mode collapse and how VS mitigates it.
    * **Human Eval & Cost Analysis:** Addressing `fHBG` and `Kajm`, we added a human study on quality (Sec 5.2), side-by-side qualitative comparisons (App H), and a detailed Cost vs. Diversity analysis (App G.2).
    * **Improved presentation and flow.** We refined the definition of typicality (Line 65), clarified computational fairness in Table 2, and added Pareto-optimal analysis (App E.1) to illustrate the diversity-quality trade-off.


3. **We believe we have effectively addressed the main concerns of the reviewers.**
    * **fHBG**
        * The concerns were about (1) Lack of direct validation on human data (2) cost and deployment feasibility analysis (3) qualitative analysis on quality.
        * We (1) improved writing clarity in Sec 3.1 and clarified that we verify the hypothesis directly on human preference data; (2) added cost/diversity trade-off analysis in App G.2; (3) conducted a human study on quality and provided additional qualitative examples in App H. We believe these additions directly addressed the reviewer's concerns that were central to their score.
    * **oDEZ:** After 1-round of rebuttal, the reviewer describe they are "nicely surprised by the results."
        * The concerns were about (1) Connection between typicality bias and Verbalized Sampling (2) missing comparable baselines (Sequence-CoT/Multi).
        * We (1) provided formal proof in Sec 4 and App D.3–D.8 that VS exploits typicality bias to restore diversity, and (2) added experiments on Sequence-CoT/Multi (on creative writing and synthetic data task). The reviewer was "nicely surprised by the results." Though we cannot receive further feedback, we believe these additions substantially address their concerns and were optimistic that they would raise their score from a 2 to a higher score.
    * **Kajm:** explicitely stated "overall I think this could be a nice paper, if the issues are resolved."
        * The concerns were about (1) regression analysis to separate what human prefer and for % of time human preferred text is typical (2) additional baselines (3) precision in writing.
        * We (1) clarified we performed their suggested regression, showing humans prefer typical responses even when utility is controlled, and added experiments demonstrating that aligned models maintain or increase typicality bias rates compared to base models; (2) added environmental randomness baselines, Pareto figures for all tasks, and moved post-training ablations on diversity to Sec 5.1; (3) improved writing clarity and scientific precision throughout. Having resolved concerns laid out by the reviewer, we are optimistic that the reviewer would have raised their score had the discussion continued.


We believe that the reviewers all found the problem we are studying to be important and of interest, and believe we have largely addressed the concerns of all reviewers, including those who could not fully engage due to the service incident.

Best regards,

The Authors

---

### Meta-Review · Area_Chair_Edc4 · 2026-01-10

**Summary:**

This paper worked mode collapse by analyzing the typicality bias in preference data with a formal definition.  Motivated by their observation, a prompting strategy Verbalized Sampling (VS) is subsequently proposed to address this issue. While the merits of the paper (e.g.,  novel formulation, simple yet effective method, comprehensive experiments) appreciated by reviewers, there are several key weaknesses that concern the reviewers (and myself). Although the authors partially addressed some of these concerns during the rebuttal, major weaknesses remain.

**Reviewer Concerns:**

After a quick examination of the paper and a careful review of all rebuttals, I found that I share the same concerns raised by the reviewers, as summarized below:
- The major concerns echoed by the reviewers  is the *Validation of Typicality Bias in Human Preference*. While the authors attempted to provide theoretical justification, empirical evidence (e.g., approximate text typicality using log probability from pretained base model, control exp), I don't think these efforts were sufficient to fully address the reviewers’ concerns (as of mine). For example, to be reasonably confident that the typicality bias been sufficiently justified, having a representative sample is crucial and it’s reasonable to expect a representative sample from as diverse preference data as (e.g possible population/demongraphic, etc)to be collected. My concern is that results from not even one representative sample are presented, and this significantly weakens the assumption.
- Further on this point, while the model’s output behaviour or generation is confounded by various variable rather than just training data (e.g., model architecture, temperature, prompting strategies, etc), the author's intuition of *as base model has been trained to maximize likelihood on massive text corpora, it will assign higher probability to more typical text* is not convincing to me.
  - A more feasible way to justify this assumption would be to start with a fully open and publicly released model (e.g., OLMo), calibrate it into a relatively controlled setting, and then conduct rigorous experiments to empirically validate this claim.
- Another concern is *Human Study on Quality*. While the author provided additional human studies on rebuttal. But still my above concern about annotator’s diversity(e.g., various background) still hold. Just a case point by Reviewer Kajm  it's unclear that they will all think the same text is typical. E.g., compare US and British english speakers.
- Finally, there are other concerns raised by the reviewers such as *missing baselines, deployment cost, qualitative analysis, writing clarity*. While the authors have addressed these points in the rebuttal, I encourage them to incorporate these analyses into the paper in subsequent revisions.

In sum, I believe this paper can become a good contribution to the community, albeit not in its current state. Addressing the issues outlined above would make the paper more impactful. and I encourage the authors to integrate these additional content into the next version of the paper as it will strengthen their contribution more.

**Reviewer Scores:**

After carefully reviewing the reviews and the rebuttal, it is possible that there maybe Reviewer increase their rating. However, I do not believe this change should be considered a decisive factor for accepting the paper. The core concerns describe above still exist. After reading the detailed rebuttal, I remain unconvinced that the authors’ responses adequately address these core issues, and thus they do not persuade me (nor, in my view, those reviewers) to revise the original scores to change the decision.

---

### Decision · Program_Chairs · 2026-01-26

Reject